# Tensor Programs I:
# Wide Feedforward or Recurrent Neural Networks of Any Architecture are Gaussian Processes

**Greg Yang**[*]
Microsoft Research AI
`gregyang@microsoft.com`

## Abstract

Wide neural networks with random weights and biases are Gaussian processes, as originally observed by Neal (1995) and more recently by Lee et al. (2018) and Matthews et al. (2018) for deep fully-connected networks, as well as by Novak et al. (2019) and Garriga-Alonso et al. (2019) for deep convolutional networks. We show that this Neural Network-Gaussian Process correspondence surprisingly extends to all modern feedforward or recurrent neural networks composed of multilayer perceptron, RNNs (e.g. LSTMs, GRUs), ($n$D or graph) convolution, pooling, skip connection, attention, batch normalization, and/or layer normalization. More generally, we introduce a language for expressing neural network computations, and our result encompasses all such expressible neural networks. This work serves as a tutorial on the *tensor programs* technique formulated in Yang (2019) and elucidates the Gaussian Process results obtained there. We provide open-source implementations of the Gaussian Process kernels of simple RNN, GRU, transformer, and batchnorm+ReLU network at `github.com/thegregyang/GP4A`.

## 1 Introduction

Motivated to understand the Bayesian prior in neural networks (NNs), Neal [41] theoretically showed that infinitely wide, shallow neural networks with random weights and biases are Gaussian processes (GPs). He empirically explored this phenomenon over deep networks as well, but this was not proven rigorously until recently [37, 40, 43, 18], with concrete progress made over the intervening years [56, 34, 22, 13]. This neural network-Gaussian process correspondence (NN-GP correspondence) has not only allowed one to transform the *implicit prior* of NNs into *explicit priors* that can be understood analytically [46, 49, 63, 59, 65], but has also created new state-of-the-art kernels by converting from deep neural networks [37, 43]. Yet, so far the focus has dwelled entirely on multilayer perceptrons (MLPs) or simple convolutional neural networks (CNNs). As new architectures are created with blistering speed, a question starts to emerge and reverberate:

*Do all infinitely wide, randomly initialized neural networks correspond to Gaussian processes?*

Even if the answer is yes, at the current rate where each new architecture warrants its own NN-GP correspondence paper, theory will never catch up to practice. On a more basic level, what does this question even mean for recurrent neural networks?

**Our Contributions** In this paper, we formulate the notion of a Gaussian process with variable-dimensional output (see Definition 2.1), and show that feedforward and recurrent neural networks of *standard architectures* converge to Gaussian processes in this sense as their widths or number

---

[*]Please see `https://arxiv.org/abs/1910.12478` for the full version of this paper.

of channels go to infinity, when their weights and biases are randomized. *By **standard architecture** we mean any architecture that is some composition of multilayer perceptrons (MLPs), recurrent neural networks (RNNs) (e.g., Long-Short Term Memory (LSTM) [26] or Gated Recurrent Unit (GRU) [10]), skip connections [24, 27], convolutions [16, 17, 47, 35, 36] or graph convolutions [8, 25, 15, 38, 14, 31], pooling [35, 36], batch normalization (batchnorm) [28], layer normalization [1] and/or attention [2, 55].* Even more broadly, we design a new language, NETSOR, for expressing neural network computations, and show the GP convergence for all such expressible networks. By demonstrating that NETSOR can implement any network of standard architectures, we obtain the aforementioned results as a corollary. The results for RNNs, batchnorm, layernorm, attention, and their combination with other layers are new. We open-source reference implementations[2] for the GP kernels of simple RNN, GRU, transformer, and feedforward batchnorm network; see Fig. 3 for an illustration.

**Relation of This Paper with [60]**   This paper serves several purposes. 1) Introduce the reader to the *tensor programs* technique formulated in [60], using the Neural Network-Gaussian Process Correspondence as motivation. 2) Promote a redesigned set of notations for *tensor programs* that hopefully makes the understanding and the application of this technique easier. 3) Prove a more general version of the Gaussian Process results first presented in [60]. 4) Provide example calculations and reference implementations[2] of the GP kernels for several architectures like the vanilla RNN, GRU, batchnorm network, and transformers.

We assume the reader has not read [60] and seek to explain all results in elementary terms. However, we will provide commentary in footnotes throughout the paper on differences from [60].

Regarding 1), this paper will be the first in a series to explain the *tensor programs* technique, each covering a more powerful type of tensor programs, and each motivated by specific theorems that can be proved or calculations made possible by these new tensor programs. In particular, here we will only talk about tensor programs without matrix transposes. Regarding 3), the results presented here will supersede all results in [60] concerning Gaussian Processes, with one caveat that here we will not cover architectures using both a weight $W$ and its transpose $W^\top$ in its forward pass (but this result will come for free in a later paper in this series).

## 2   Gaussian Process with Variable-Dimensional Output

We first clarify the notion of a Gaussian process with variable dimension output.

**Definition 2.1** (Gaussian Process).  We say a random function $f : X \to \mathbb{R}^m$ (with fixed dimensional output) is a Gaussian process if for any finite subset $\{x^1, \dots, x^k\} \subseteq X$, the random vector $(f(x^1), \dots, f(x^k)) \in \mathbb{R}^{m \times k}$ is distributed as a $km$-dimensional Gaussian. If $f$ has variable dimensional output (e.g. $f$ is an RNN), such as when $f(x) \in \mathbb{R}^{l(x)}$ for some length function $l : X \to \mathbb{N}$ [3], then we say $f$ is a Gaussian process if for any finite subset $\{x^1, \dots, x^k\} \subseteq X$, the random vector $(f(x^1), \dots, f(x^k))$ is distributed as a $(\sum_i l(x^i))$-dimensional Gaussian.

To illustrate a GP with variable-dimensional output, consider a simple RNN that runs on two input sequences given by the GloVe embeddings [44] [4] of the words of the two sentences

$$\begin{aligned}&\text{sentence 1 (7 words):} \quad \text{``The brown fox jumps over the dog.''}\\&\text{sentence 2 (9 words):} \quad \text{``The quick brown fox jumps over the lazy dog.''}\end{aligned} \qquad (\star)$$

A pseudocode is given in Program 2 in Section 4 (ignore the type annotations like $\mathsf{G}(n), \mathsf{H}(n), \mathsf{A}(n)$ for now). The RNN emits a single scalar after reading each token (in Program 2, this is $v^\top s^{ia}/\sqrt{n}$, where $s^{ia}$ is the RNN state after reading the $i$th token of the $a$th sentence, and $v$ is the readout layer); this number takes into account all of the word embeddings read so far. Thus, it will output a total of 7 scalars after reading sentence 1, and a total of 9 scalars after reading sentence 2. To say that this RNN is a GP would imply that all $7 + 9 = 16$ scalars are jointly Gaussian-distributed (corresponding to a $16 \times 16$ kernel), over the randomness of the weights and biases imbued during initialization. This

is indeed the empirical phenomenon with a width-1000 RNN, and Fig. 2(E) visualizes the the joint distribution of the last scalars output by the RNN at the end of each sentence. It clearly exhibits a Gaussian nature, and perfectly fits the theoretically predicted Gaussian distribution (dashed ovals), which we shall describe in Corollary 5.5.

## 3 Recap: GP Behavior of a Multilayer Perceptron (MLP)

Before explaining our main results, we first review the argument from prior works [37, 40, 43] for the GP convergence of a wide MLP with randomly initialized weights and biases, and we also demonstrate why such an argument is inadequate for RNNs. Consider an MLP with widths $\{n^l\}_l$, weight matrices $\{W^l \in \mathbb{R}^{n^l \times n^{l-1}}\}_l$, and biases $\{b^l \in \mathbb{R}^{n^l}\}_l$, where $l$ ranges among the layer numbers of the MLP. Its computation is given recursively as

$$h^1(x) = W^1 x + b^1 \qquad \text{and} \qquad h^l(x) = W^l \phi(h^{l-1}(x)) + b^l \text{ for } l \geq 2. \tag{1}$$

At initialization time, suppose $W^l_{\alpha\beta} \sim \mathcal{N}(0, \sigma_w^2/n^{l-1})$ for each $\alpha \in [n^l], \beta \in [n^{l-1}]$, and $b^l_\alpha \sim \mathcal{N}(0, \sigma_b^2)$. Consider two inputs $x, x'$. Conditioned on $h^{l-1}(x)$ and $h^{l-1}(x')$, iid for each $\alpha, (h^l(x)_\alpha, h^l(x')_\alpha)$ is distributed as

$$\mathcal{N}\left(0, \frac{\sigma_w^2}{n^{l-1}} \begin{pmatrix} \|\phi(h^{l-1}(x))\|^2 & \phi(h^{l-1}(x)) \cdot \phi(h^{l-1}(x')) \\ \phi(h^{l-1}(x)) \cdot \phi(h^{l-1}(x')) & \|\phi(h^{l-1}(x'))\|^2 \end{pmatrix} + \sigma_b^2 . \right)$$

If $(h^{l-1}(x)_\alpha, h^{l-1}(x')_\alpha)$ is distributed as $\mathcal{N}(0, \Sigma^{l-1})$, iid for each $\alpha$, then by a law of large number argument, the covariance matrix above converges to a deterministic limit

$$\Sigma^l \stackrel{\text{def}}{=} \sigma_w^2 \mathop{\mathbb{E}}_{(z,z') \sim \mathcal{N}(0, \Sigma^{l-1})} \begin{pmatrix} \phi(z)^2 & \phi(z)\phi(z') \\ \phi(z)\phi(z') & \phi(z')^2 \end{pmatrix} + \sigma_b^2$$

as the width $n^{l-1} \to \infty$, making $(h^l(x)_\alpha, h^l(x')_\alpha)$ Gaussian distributed as $\mathcal{N}(0, \Sigma^l)$. Iteratively applying this argument for each $l$ yields the result for a deep MLP. A similar logic works for feedforward CNNs.

Unfortunately, this argument breaks down if the weights $\{W^l\}_l$ are tied, i.e. all $W^l$ are equal to a common matrix $W$, as in the case of an RNN. In this case, when we condition on the preactivations $h^{l-1}(x), h^{l-1}(x')$ of the previous layer, $W$ is no longer conditionally an iid random Gaussian matrix, and all subsequent reasoning breaks down. We can repair this situation for RNNs in an ad hoc way via the Gaussian conditioning technique (Lemma G.7), but we prefer to set our sights wider, and deal with all standard architectures, and more, in one fell swoop. To this end, we develop a framework based on our new NETSOR language.

## 4 NETSOR : Language for Expressing Neural Network Computation

To show that networks of all standard architectures converge to GPs, we first show that they can be expressed by the following very general NETSOR language (see Programs 1 and 2 for examples)[5], and then show that any computation expressed this way exhibits GP behavior when its dimensions are large.

**Definition 4.1.** [6] NETSOR *programs* are straight-line programs, where each variable follows one of three types, G, H, or A (such variables are called *G-vars*, *H-vars*, and *A-vars*), and after input variables, new variables can be introduced by one of the rules `MatMul`, `LinComb`, `Nonlin` to be discussed shortly. G and H are *vector types* and A is a *matrix type*; intuitively, G-vars should be thought of as vectors that are asymptotically Gaussian, H-vars are images of G-vars by coordinatewise nonlinearities, and A-vars are random matrices with iid Gaussian entries. Each type is annotated by dimensionality information:

- If $x$ is a (vector) variable of type G (or H) and has dimension $n$, we write $x : \mathsf{G}(n)$ (or $x : \mathsf{H}(n)$).

**NETSOR program 1** MLP Computation on Network Input $x$
---
**Input:** $W^1 x : \mathsf{G}(n^1)$         ▷ layer 1 embedding of input
**Input:** $b^1 : \mathsf{G}(n^1)$         ▷ layer 1 bias
**Input:** $W^2 : \mathsf{A}(n^2, n^1)$         ▷ layer 2 weights
**Input:** $b^2 : \mathsf{G}(n^2)$         ▷ layer 2 bias
**Input:** $v : \mathsf{G}(n^2)$         ▷ readout layer weights
  1: $h^1 := W^1 x + b^1 : \mathsf{G}(n^1)$         ▷ layer 1 preactivation; `LinComb`
  2: $x^1 := \phi(h^1) : \mathsf{H}(n^1)$         ▷ layer 1 activation; `Nonlin`
  3: $\tilde{h}^2 := W^2 x^1 : \mathsf{G}(n^2)$         ▷ `MatMul`
  4: $h^2 := \tilde{h}^2 + b^2 : \mathsf{G}(n^2)$         ▷ layer 2 preactivation; `LinComb`
  5: $x^2 := \phi(h^2) : \mathsf{H}(n^2)$         ▷ layer 2 activation; `Nonlin`
**Output:** $v^\top x^2 / \sqrt{n^2}$
---

- If $A$ is a (matrix) variable of type $\mathsf{A}$ and has size $n_1 \times n_2$, we write $A : \mathsf{A}(n_1, n_2)$.

$\mathsf{G}$ is a *subtype* of $\mathsf{H}$, so that $x : \mathsf{G}(n)$ implies $x : \mathsf{H}(n)$. A NETSOR program consists of the following three parts.

**Input** A set of input $\mathsf{G}$- or $\mathsf{A}$-vars.

**Body** New variables can be introduced and assigned via the following rules (with *intuition in italics*)

> `MatMul` if $A : \mathsf{A}(n_1, n_2)$ and $x : \mathsf{H}(n_2)$, we can form a $\mathsf{G}$-var via matrix-vector product:
>
>      $Ax : \mathsf{G}(n_1)$,    *"random iid matrix times a vector is roughly a Gaussian vector."*[7]

> `LinComb` Suppose $x^1, \ldots, x^k : \mathsf{G}(n)$ are $\mathsf{G}$-vars with the same dimension and $a_1, \ldots a_k \in \mathbb{R}$ are constants. Then we can form their linear combination as a $\mathsf{G}$-var:
>
> $$\sum_{i=1}^{n} a_i x^i : \mathsf{G}(n), \quad \text{"linear combination of Gaussian vectors is Gaussian."}$$

> `Nonlin` If $x^1, \ldots, x^k : \mathsf{G}(n)$ are $\mathsf{G}$-vars with the same dimension $n$ and $\phi : \mathbb{R}^k \to \mathbb{R}$, then
>
>      $\phi(x^1, \ldots, x^k) : \mathsf{H}(n)$,    *"image of Gaussian vector is not always Gaussian"*
>
> where $\phi$ acts coordinatewise.

**Output** For the purpose of this paper[8], the output of a NETSOR program can be any tuple of scalars, $(v^{1\top} y^1 / \sqrt{n_1}, \ldots, v^{k\top} y^k / \sqrt{n_k})$, where $v^1 : \mathsf{G}(n_1); \ldots; v^k : \mathsf{G}(n_k)$ are some input $\mathsf{G}$-vars not used elsewhere (and possibly with duplicates $v^i = v^j$), and $y^1 : \mathsf{H}(n_1); \ldots; y^k : \mathsf{H}(n_k)$ are some $\mathsf{H}$-vars (possibly with duplicates $y^i = y^j$).

**Examples** Program 1 gives an example of a NETSOR program representing an MLP computation. Note that *we account for the input $x$ through its embedding $W^1 x$, not $x$ itself.* This is because 1) our theorems concern the case where all input $\mathsf{G}$-vars are random; in the context of expressing neural network computation, $x$ is a deterministic input, while $W^1 x$ is a Gaussian vector when $W^1$ has iid Gaussian entries; 2) $x$ has a fixed dimension, while we intend all dimensions (like $n^1, n^2$) in the NETSOR program to tend to infinity, as we'll describe shortly. Similarly, Program 2 expresses in NETSOR the computation of a simple RNN on two separate input sequences; computation on more input sequences follows the same pattern. Note how weight-sharing is easily expressed in NETSOR because we can re-use $\mathsf{A}$-vars arbitrarily. Appendix A shows more examples of standard architectures in NETSOR and NETSOR$^+$ .

More generally, we can allow the nonlinearities in `Nonlin` to depend on parameters; this will be necessary to express layernorm and attention (see Appendix A). We capture this idea in a new rule:

**NETSOR program 2** Simple RNN Computation on Two Input Sequences

---

*// Embeddings of two inputs sequences*
**Input:** $Ux^{11}, \ldots, Ux^{t1} : \mathsf{G}(n)$
**Input:** $Ux^{12}, \ldots, Ux^{r2} : \mathsf{G}(n)$
*// Weight and bias*
**Input:** $W : \mathsf{A}(n, n)$
**Input:** $b : \mathsf{G}(n)$
*// Readout weights*
**Input:** $v : \mathsf{G}(n)$
*// Computation on sequence 1*
$h^{11} := Ux^{11} + b : \mathsf{G}(n)$
$s^{11} := \phi(h^{11}) : \mathsf{H}(n)$
$\tilde{h}^{21} := Ws^{11} : \mathsf{G}(n)$
$h^{21} := \tilde{h}^{21} + Ux^{21} + b : \mathsf{G}(n)$
$s^{21} := \phi(h^{21}) : \mathsf{H}(n)$
$\vdots$

$\tilde{h}^{t1} := Ws^{t-1,1} : \mathsf{G}(n)$
$h^{t1} := \tilde{h}^{t1} + Ux^{t1} + b : \mathsf{G}(n)$
$s^{t1} := \phi(h^{t1}) : \mathsf{H}(n)$
*// Computation on sequence 2*
$h^{12} := Ux^{12} + b : \mathsf{G}(n)$
$s^{12} := \phi(h^{12}) : \mathsf{H}(n)$
$\tilde{h}^{22} := Ws^{12} : \mathsf{G}(n)$
$h^{22} := \tilde{h}^{22} + Ux^{22} + b : \mathsf{G}(n)$
$s^{22} := \phi(h^{22}) : \mathsf{H}(n)$
$\vdots$

$\tilde{h}^{r2} := Ws^{r-1,2} : \mathsf{G}(n)$
$h^{r2} := \tilde{h}^{r2} + Ux^{r2} + b : \mathsf{G}(n)$
$s^{r2} := \phi(h^{r2}) : \mathsf{H}(n)$
**Output:** $(v^\top s^{11}/\sqrt{n}, \ldots, v^\top s^{t1}/\sqrt{n},$
$\qquad\qquad v^\top s^{12}/\sqrt{n}, \ldots, v^\top s^{r2}/\sqrt{n})$

---

$\texttt{Nonlin}^+$  Suppose $x^1, \ldots, x^k : \mathsf{G}(n)$ are G-vars with the same dimension $n$ and $\theta_1, \ldots, \theta_t \in \mathbb{R}$ possibly depend on G-vars already defined. If $\phi(-;-) : \mathbb{R}^k \times \mathbb{R}^t \to \mathbb{R}$, then

$$\phi(x^1, \ldots, x^k; \theta_1, \ldots, \theta_t) : \mathsf{H}(n),$$

where $\phi$ acts coordinatewise.

**Definition 4.2.** NETSOR$^+$ programs are NETSOR programs allowing $\texttt{Nonlin}^+$ rules.

NETSOR and NETSOR$^+$ specify different kinds of *tensor programs*; in Appendix E we discuss other kinds that are semantically equivalent. In a future paper, we shall study the effect of allowing matrix transposes as an operation on A-vars.

*Remark* 4.3. In this paper, in $\texttt{Nonlin}^+$, we will only instantiate $\theta_j$ with continuous functions of "empirical moments" of the form $n^{-1} \sum_{i=1}^n \psi(y^1, \ldots, y^r)$ for some set of G-vars $\{y_i\}_i$. A key consequence of our scaling limit result is that these "empirical moments" converge almost surely to a deterministic limit under very general conditions (Theorems 5.4 and C.4), so that $\phi(-;\Theta)$ is, under suitable smoothness conditions (Definition C.1), approximately a fixed nonlinearity when $n$ is large. Thus, we should intuitively treat $\texttt{Nonlin}^+$ as $\texttt{Nonlin}$ but with the nonlinearity determined automatically by the NETSOR program itself.

$\texttt{Nonlin}^+$ expands the expressible computation quite broadly, but to keep the main text lean and focused on the key ideas behind tensor programs, we relegate a more thorough discussion of $\texttt{Nonlin}^+$ in the appendix (see Appendices C, D and I).

## 5  Computing the GP Kernel from a NETSOR Encoding of a Neural Network

For readers who wish to be convinced that NETSOR (or NETSOR$^+$ ) can express standard architectures, see Appendix A. In this section, we show that any architecture expressible in NETSOR and satisfies some mild conditions will exhibit Gaussian Process behavior in the large width limit.

In this section, we make the following simplifying assumption on the dimensions of the program and the randomness of the variables.

**Assumption 5.1.** *Fix a* NETSOR *program. For simplicity, assume all dimensions in the program are equal to $n$. Suppose for each A-var $W : \mathsf{A}(n, n)$, we sample $W_{\alpha\beta} \sim \mathcal{N}(0, \sigma_W^2/n)$ for some $\sigma_W^2 > 0$, and for each $\alpha \in [n]$, we sample, i.i.d., $\{x_\alpha : x$ is input G-var$\} \sim \mathcal{N}(\mu^{\text{in}}, \Sigma^{\text{in}})$ for some mean $\mu^{\text{in}}$ and (possibly singular) covariance $\Sigma^{\text{in}}$ over input G-vars.*

The constraint on the dimensions can be removed easily; see Appendix F. This sampling induces randomness in all variables created in the program, and we shall characterize this randomness shortly. We first review some notation that will be used immediately.

**Notation**   In this paper, a *kernel* $\Sigma$ *on a set* $X$ is a symmetric function $\Sigma : X \times X \to \mathbb{R}$ such that

$$\sum_{i=1}^{m} \sum_{j=1}^{m} c_i c_j \Sigma(x_i, x_j) \geq 0$$

holds for any $m \in \mathbb{N}$, $x_1, \ldots, x_m \in X$, and $c_1, \ldots, c_m \in X$. Given a kernel $\Sigma$ on a set of G-vars, we will both treat it as matrix and as a function, depending on the context. **Function Notation**   As a function, $\Sigma(g, g')$ is the value of $\Sigma$ on the pair of G-vars $(g, g')$. If $G = \{g^1, \ldots, g^k\}$ is a set of G-vars, then we also denote by $\Sigma(g, G)$ the row vector $\{\Sigma(g, g^1), \ldots, \Sigma(g, g^k)\}$. Likewise $\Sigma(G, g)$ is the column vector with the same values. If $G' = \{g^{1'}, \ldots, g^{l'}\}$ is another set of G-vars (possible with overlap with $G$), then $\Sigma(G, G')$ is the matrix $\{\Sigma(g^i, g^{j'}) : i \in [k], j \in [l]\}$. **Restriction Notation** We also use the "restriction" notation $\Sigma|_G$ to denote the square matrix $\Sigma(G, G)$ in a more concise way. **Matrix Notation**   When an association of indices to G-vars is clear from context, we will also write $\Sigma_{ij}$ for the corresponding value of $\Sigma$ on the pair of $i$th and $j$th G-vars. Juxtaposition implies matrix multiplication, e.g. $\Sigma\Omega$ means matrix product if $\Omega$ is a matrix of appropriate size. **Indices Notation** We will both use superscripts and subscripts for indices. We will never multiply in subscript or superscript, so juxtaposition of indices like $W_{\alpha\beta}^{ib}$ is the same as $W_{\alpha,\beta}^{i,b}$. **H-vars as Both Symbols and Vectors** An H-var will be considered both as a symbol (like in $\Sigma(g, g')$ above) as well as the corresponding length $n$ vector (like in Theorem 5.4 below), depending on the context.

**Definition 5.2.**   In the setting of Assumption 5.1, we extend $\mu^{\text{in}}$ and $\Sigma^{\text{in}}$ to $\mu$ and $\Sigma$ that resp. take a single and a pair of G-vars and both output to $\mathbb{R}$. Intuitively, $\mu$ specifies the mean coordinate of a G-var, and $\Sigma$ specifies the coordinatewise covariance of a pair of G-vars; this is formalized in Theorem 5.4 below. Index all the G-vars in the program as $g^1, \ldots, g^M$ (including input G-vars), in the order of appearance in the program. For any pair of G-vars $g, g'$ (among $g^1, \ldots, g^M$), we define recursively

$$\mu(g) = \begin{cases} \mu^{\text{in}}(g) & \text{if } g \text{ is input} \\ \sum_i a_i \mu(y^i) & \text{if } g = \sum_i a_i y^i, \text{introduced by } \texttt{LinComb} , \\ 0 & \text{otherwise} \end{cases}$$

$$\Sigma(g, g') = \begin{cases} \Sigma^{\text{in}}(g, g') & \text{if } g, g' \text{ are inputs} \\ \sum_i a_i \Sigma(y^i, g') & \text{if } g = \sum_i a_i y^i, \text{introduced by } \texttt{LinComb} \\ \sum_i a_i \Sigma(g, y^i) & \text{if } g' = \sum_i a_i y^i, \text{introduced by } \texttt{LinComb} \\ \sigma_W^2 \, \mathbb{E}_Z \, \phi(Z)\bar{\phi}(Z) & \text{if } g = Wh, g' = Wh', \text{introduced by } \texttt{MatMul} \text{ w/ same A-var } W \\ 0 & \text{otherwise} \end{cases}$$

(2)

where

- $y^i$ are G-vars for all $i$

- $(h : \text{H}(n))$ was introduced by the `Nonlin` with $h := \phi(g^1, \ldots, g^M)$, $h'$ was introduced by `Nonlin` with $h' := \bar{\phi}(g^1, \ldots, g^M)$ (where WLOG we have padded the input slots of $\phi$ and $\bar{\phi}$ to account for all G-vars)

- $Z \sim \mathcal{N}(\mu, \Sigma)$ is a random Gaussian vector with 1 entry for each G-var in the program.

Note that since $\phi$ and $\bar{\phi}$ only depends on entries of $Z$ corresponding to previous G-vars, the expectation $\mathbb{E}_Z \, \phi(Z)\bar{\phi}(Z)$ only depends on entries of $\mu$ and $\Sigma$ already defined, so there is no circular logic in this recursive definition of $\mu$ and $\Sigma$. See Appendix B.1.1 for a simple, worked-out example of how to recursively compute $\mu$ and $\Sigma$ for Program 1.

For our main theorems, we isolate a very general class of nonlinearities that we are concerned with.

**Definition 5.3.**   We say a function $\phi : \mathbb{R}^k \to \mathbb{R}$ is *controlled* if $|\phi(x)|$ is bounded by a function of the form $e^{C\|x\|^{2-\epsilon}+c}$ with $C, c, \epsilon > 0$

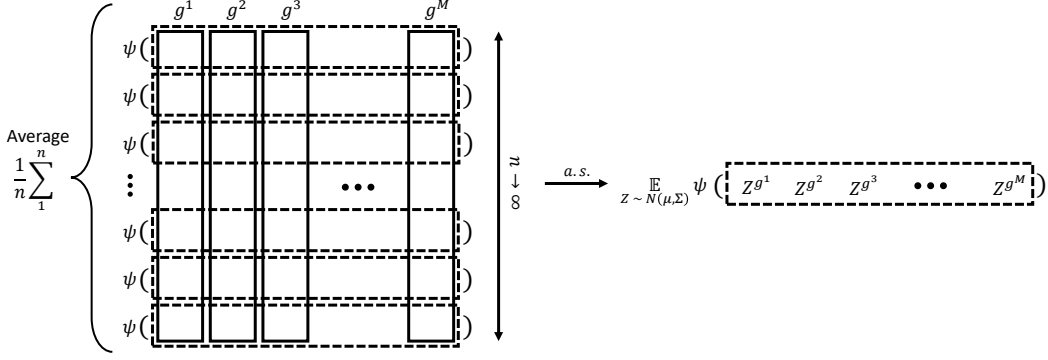

Figure 1: An illustration of the NETSOR Master Theorem Theorem 5.4.

Controlled functions can explode faster than exponential but are still $L^1$ and $L^2$-integrable against Gaussian measures. Additionally, there is no constraint on the smoothness of $\phi$ here. Thus this definition captures almost all functions we would care about in practice.

The metric structure of the final layer representations of inputs under a deep neural network often reveals semantical information about the inputs. This structure is reflected in the inner products between pairs of such representations, e.g. $s^{t1\top}s^{r2}/n$ for $s^{t1}$ and $s^{r2}$ in Program 2. The following Master Theorem allows one to compute such inner products, and much more, for a wide network at initialization time (take $\psi$ below to be $\psi(z^1,\dots,z^M) \overset{\text{def}}{=} z^{M-1}z^M$).

**Theorem 5.4** (NETSOR Master Theorem). [9] *Fix any NETSOR program satisfying Assumption 5.1 and with all nonlinearities controlled. If $g^1,\dots,g^M$ are all of the G-vars in the entire program, including all input G-vars, then for any controlled $\psi : \mathbb{R}^M \to \mathbb{R}$, as $n \to \infty$,*

$$\frac{1}{n}\sum_{\alpha=1}^{n} \psi(g^1_\alpha,\dots,g^M_\alpha) \xrightarrow{\text{a.s.}} \underset{Z\sim\mathcal{N}(\mu,\Sigma)}{\mathbb{E}} \psi(Z) = \underset{Z\sim\mathcal{N}(\mu,\Sigma)}{\mathbb{E}} \psi(Z^{g^1},\dots,Z^{g^M}),$$

*where $\xrightarrow{\text{a.s.}}$ means almost sure convergence, $Z = (Z^{g^1},\dots,Z^{g^M}) \in \mathbb{R}^M$, and $\mu = \{\mu(g^i)\}_{i=1}^M \in \mathbb{R}^M$ and $\Sigma = \{\Sigma(g^i,g^j)\}_{i,j=1}^M \in \mathbb{R}^{M\times M}$ are given in Eq. (2). See Fig. 1 for an illustration.*

Intuitively, Theorem 5.4 says, for each $\alpha$, $(g^1_\alpha,\dots,g^M_\alpha) \approx \mathcal{N}(\mu,\Sigma)$ in the large $n$ limit, and each $\alpha$-slice appears to be "iid" from the point of view of the empirical average by any controlled function $\psi$. The proof of Theorem 5.4 is given in Appendix H.

Combining Theorem 5.4 with Proposition G.4, we can straightforwardly calculate the output distribution of a NETSOR program.

**Corollary 5.5** (Computing the GP Kernel). *Adopt the same assumptions and notations as in Theorem 5.4. If the program outputs $(v^\top x^1/\sqrt{n},\dots,v^\top x^k/\sqrt{n})$, where*

- *$v : \mathsf{G}(n), v_\alpha \sim \mathcal{N}(0,\sigma_v^2)$, is an input G-var not used elsewhere in the program and is sampled independently from all other G-vars, and*

- *$x^i$ was introduced as $x^i := \phi^i(g^1,\dots,g^M)$*

*then the output vector converges in distribution to $\mathcal{N}(0,K)$ where*

$$K_{ij} = \sigma_v^2 \underset{Z\sim\mathcal{N}(\mu,\Sigma)}{\mathbb{E}} \phi^i(Z)\phi^j(Z), \quad \text{with } \mu,\Sigma \text{ defined in Eq. (2).} \tag{3}$$

Intuitively, this corollary follows from the fact that, for any finite $n$, the output vector is some Gaussian $\mathcal{N}(0,\tilde{K})$ conditioned on $x^1,\dots,x^k$, and the covariance $\tilde{K}$ converges to a deterministic covariance $K$, causing the output vector to converge in distribution to $\mathcal{N}(0,K)$ as well. The case

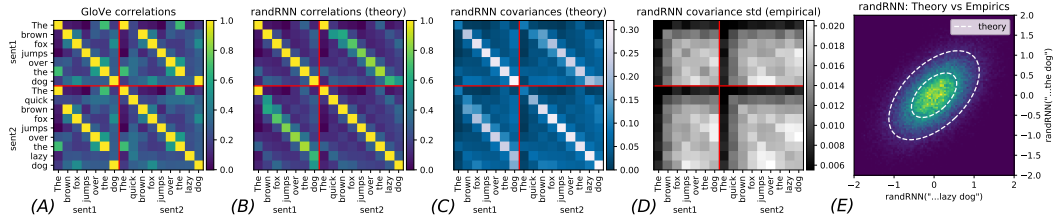

Figure 2: *Infinite-width theory is highly predictive for simple RNN (Program 2) with 1000 neurons and erf activation.* We pass two sentences ("The brown fox jumps over the dog" and "The quick brown fox jumps over the lazy dog") by their word GloVe embeddings into randomly initialized simple RNNs. **(A)** Cosine distances between each pair of word GloVe embeddings. **(B)** Correlation matrix of the limiting Gaussian that Program 2 output vector converges to. Each row/column corresponds to an embedding of of the sentence up to that word. **(C)** *Covariance* matrix of the same. See Appendix B.2 for algorithm to compute this covariance. **(D)** Entrywise standard deviation of empirical covariance around (C), as measured from 100 random simple RNNs. Note the deviations are at least an order of magnitude smaller than the limiting values (C), for 1000 neurons. **(E)** Visualizing the joint distribution of the final outputs of the RNN at the end of each sentence, i.e. $(v^\top s^{t1}/\sqrt{n}, v^\top s^{r2}/\sqrt{n})$ in Program 2. We sampled 100,000 simple RNNs and plotted the 2d histogram as a heatmap. Simultaneously, we plot the limiting Gaussian level curves predicted by our theory, which fit the simulations perfectly.

when we have multiple distinct $v^i$ (allowed by Definition 4.1) can be obtained easily as well (see Proposition G.4).

Following Corollary 5.5 and its extensions below, the convergence of standard architectures to Gaussian Processes becomes obvious: Express the marginal of the distribution on every finite set of inputs as a NETSOR (or NETSOR$^+$ ) program, and then apply Corollary 5.5. We summarize the result below.

**Corollary 5.6.** *If its nonlinearities are controlled (Definition 5.3), then a (possibly recurrent) neural network of standard architecture converges to a Gaussian process in finite-dimensional distribution [10] in the sense of Definition 2.1 as its widths go to infinity and each of its weights $W$ and biases $b$ are randomized as $W_{\alpha\beta} \sim \mathcal{N}(0, \sigma_W^2/n), b_\alpha \sim \mathcal{N}(\mu_b, \sigma_b^2)$ for a collection of sampling hyperparameters $\{\sigma_W\}_W, \{\mu_b, \sigma_b\}_b$. If its nonlinearities are more generally parametrized and are parameter-controlled (Definition C.1), such as in the case of attention models or where layernorm is involved, then the same result holds as long as Assumption C.3 also holds.*

**An Empirical Demonstration**   Despite being about infinite width, our theory is highly predictive for finite-width networks, as shown in Fig. 2. As in Section 2, we randomly initialize a simple RNN (Program 2) with 1000 neurons and erf activation (we choose erf instead of tanh because it simplifies kernel calculations; see Appendix B.2 for the derivation of the algorithm to compute the kernel). We pass the two sentences in (⋆) to the random RNN by their GloVe embeddings. After processing each token, the RNN outputs a scalar, as before, and over the two input sequences, the RNN outputs $7 + 9 = 16$ scalars in total. Our result Corollary 5.5 implies that, as the width of the RNN grows to infinity, these 16 scalars are distributed jointly as a Gaussian. Fig. 2(E) illustrates this is indeed the case for the marginal on 2 scalars, as discussed in Section 2. We also compare our theoretically derived, infinite-width covariance of the 16 scalars (Fig. 2(C)) with the empirical finite-width covariance obtained from multiple random initializations. We find that the empirical covariance, as predicted, concentrates around the theoretical, and the entrywise standard deviation is typically at least an order of magnitude lower than the values themselves (Fig. 2(D)) (with width 1000 RNNs). The random RNN is clearly doing nontrivial context embedding, as seen by comparing the *correlation* matrix of the 16 scalars Fig. 2(B) (context-sensitive) with the matrix of cosine distances (i.e. correlations) between the GloVe embeddings Fig. 2(A) (context-insensitive). A tell-tale sign is the entry corresponding to ("lazy", "dog"): even though as words, they are not semantically similar

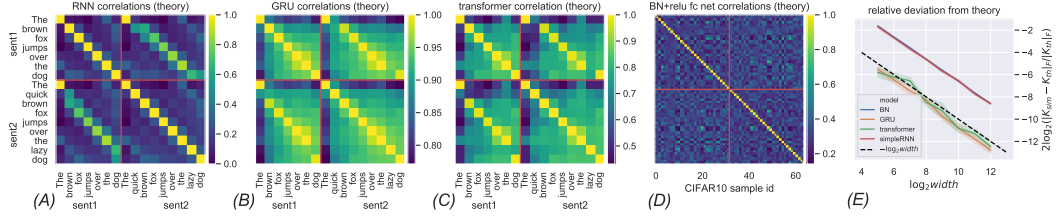

Figure 3: *Infinite-width GP kernels (more precisely, their correlation matrices) for which we provide reference implementations, and the deviation of finite-width simulations from the corresponding infinite-width limits.* **(A) – (C)** The correlation matrices of the GP kernels for the simple RNN (same as in Fig. 2; see Program 2 for the architecture and Appendix B.2 for derivation), GRU (Program 5; Appendix B.5), and transformer (Program 10; Appendix D.3), with input sequences given by the GloVe embeddings of (⋆). **(D)** The correlation matrix of the GP kernel for a feedforward, fully-connected network with batchnorm+ReLU (batchnorm followed by ReLU) as nonlinearity (see Appendix B.3 for derivation). The inputs are the first 64 CIFAR10 images, split into two batches of 32 each. The red lines indicate the batch split. **(E)** For each architecture above, we independently randomly initialize 100 networks for each width among $[2^5, 2^6, \dots, 2^{13}]$. We calculate the empirical kernel of the network outputs and plot its (relative) Frobenius distance to the infinite-width kernel. This Frobenius distance drops like $1/\sqrt{width}$ as one would expect from a central limit intuition. See our code[2] for Python implementations of these kernels and the code for generating this figure.

(so that the entry in Fig. 2(A) is small), the random RNN understands that the two sentences resp. up to "lazy" and "dog" have been very similar (so that the entry in Fig. 2(B) is large). Given the precision of our theoretical predictions, we expect analyses of the equations derived here will lead to many nontrivial insights about recurrent (and other) neural network behavior in practice, which we leave for future work.

**Examples and Extensions: A Brief Guide to the Appendix** Appendix B contains a plethora of worked-out examples of the kernel computation for different architectures, starting from the known case of MLP to the new results of RNN (as shown in Fig. 2), GRU, batchnorm, and others. At this point, we recommend the reader to follow along some of those examples to solidify the understanding of Theorem 5.4.

A Master Theorem for NETSOR$^+$ can be similarly proved. This is stated in Appendix C and can be proved easily given the proof of Theorem 5.4; see Appendix I. Appendix D works out examples of kernel computations for layernorm and transformer, which can only be expressed through NETSOR$^+$. Fig. 3 illustrates the kernels of simple RNN, GRU, transformer, and a batchnorm+ReLU network, and confirms that the finite width simulations tend to the infinite-width, theoretical kernels.

We also discuss different variants of NETSOR and NETSOR$^+$ in Appendix E which trade off syntactical simplicity with ease of use, but are semantically equivalent to NETSOR or NETSOR$^+$. Appendix F discusses the case when the dimensions of a program need not be equal. With the appropriate setup, a Master Theorem in this case can be proved similarly (Theorem F.4).

## 6 Related Works

**NN-GP Correspondence** Many works have observed the neural network-Gaussian process correspondence (NN-GP correspondence) for subsets of standard architectures [56, 34, 22, 13, 37, 40, 43]. Others have exploited this NN-GP correspondence implicitly or explicitly to build new models [11, 33, 12, 57, 58, 7, 54, 32, 4, 6, 18, 43]. In particular, by directly converting NN to GP using this correspondence, Lee et al. [37] constructed the state-of-the-art (SOTA) permutation-invariant GP on MNIST, and Novak et al. [43] was until recently SOTA on CIFAR10 for any GP with untrainable kernel. Additionally, the NN-GP correspondence has led to new understanding of neural network training and generalization [42, 53, 61].

In this paper, we generalized the NN-GP correspondence to *standard architectures* and very general nonlinearities (controlled functions; see Definition 5.3). In contrast, Matthews et al. [40] requires $\phi$ to be linearly bounded in norm; Daniely et al. [13] requires $\phi$ be twice-differentiable with $|\phi|, |\phi'|, |\phi''|$

all bounded, or that $\phi = \text{ReLU}$; and a sufficient condition given in Novak et al. [43] is that $\phi'$ exists and is bounded by $\exp(O(x^{2-\epsilon}))$, though it is unclear how the more general set of 3 conditions given there (in their section E.4) compares with ours.

**Signal Propagation in Neural Networks** A long line of work starting with Glorot and Bengio [20] and He et al. [23] studies the effect of initialization in deep neural networks [46, 50, 63, 62, 21, 9, 64, 45], for example, what is the best initialization scheme to avoid gradient vanishing? These works apply the same calculations of covariances as we do for calculating $\Sigma$ here, though in a much more restricted set of architectures, and they are typically more concerned with the dynamics of such covariances with depth.

**Reservoir Computing** In reservoir computing [30, 39, 51], sequence processing is typically done by a randomly initialized recurrent neural network. A sequence of inputs is fed step by step into the network, and a final readout layer transforms the random RNN's state into an output. The only trainable parameters are the readout layer, but not the random RNN itself. Thus, in the infinite-width limit, reservoir computing corresponds exactly to GP inference with the RNN kernel computed in Appendix B.2.

## 7   Conclusion

We formulated the notion of Gaussian process with variable-dimensional outputs and showed that randomly initialized, wide feedforward and recurrent neural networks of standard architectures converge in distribution to Gaussian processes in such a sense. This significantly generalizes prior work on the NN-GP correspondence. We did so by introducing NETSOR, a language for expressing computation common in deep learning, including neural networks of standard architecture, along with a theorem (Theorem 5.4) characterizing the behavior of a NETSOR program as its tensors are randomized and their dimensions tend to infinity; many examples and extensions are exhibited in the appendix. Finally, we empirically verified our theory for simple RNN, GRU, transformer, and batchnorm (Fig. 3) and open-sourced implementations of the corresponding infinite-width limit kernels at `github.com/thegregyang/GP4A`. In the next paper in this series, we will introduce a more powerful version of tensor program that allows matrix transposes, and use this tool to compute Neural Tangent Kernel [29] for any architecture.

## Footnotes

[2] github.com/thegregyang/GP4A

[3] i.e. $f : \prod_{x \in X} \mathbb{R}^{l(x)}$ is a dependent function

[4] The embedding associates each word to a real vector of 100 dimensions such that semantically similar words are mapped to closer vectors

[5]NETSOR is a specific kind of tensor program; for other variants, see Appendix E.

[6]We keep the definition here informal in terms of programming language convention to be accessible to the general machine learning audience. For those with PL background, see Appendix J.

[7]Beware: in a later paper (and in [60], tensor program general case), we will introduce matrix transpose as a valid operation, and in that case, $Ax$ can be very far from a Gaussian, and this intuition is no longer correct. Thus, this intuition is more subtle than it might seem at face value.

[8]In general, the output of a tensor program need not be defined, as most of the time we are concerned with how the $\mathsf{H}$-vars produced over the course of the program interact with each other.

[9]Difference with [60, Thm 4.3]: We have gotten rid of the "rank convergence" assumption by showing that it comes for free. See CoreSet and Lemma H.4 in Appendix H.

[10]Stronger convergence results, such as convergence in distribution with respect to some topology on functions on $\mathbb{R}^d$, would be available if one can show additionally the *tightness* of the random neural networks under this topology. However, here we are content with convergence of finite-dimensional marginals of the stochastic processes.
