[Reviews · NeurIPS 2019]

Reviewer 1



This review has 2 parts. The first part is my review of the paper as a standalone paper. The second part is a meta-commentary unifying my reviews for both this paper and "Neural Tangent Kernel for Any Architecture". ++++++++ Part 1 ++++++++++++++ This paper demonstrates that infinitely-wide architectures made from a range of building blocks are Gaussian processes. Fundamentally, the paper seems to have two core contributions. (1) The paper collapses a wide range of operations (convolution, pooling, batchnorm, attention, gating, as well as the inner products for the actual GP Kernel computation) into the matrix multiplication / nonlinearity / linear combination framework. (2) The paper presents a mean field theory of tied weights, which allows a rigorous extension to RNNs as well as a rigorous integration of the forward and backward pass. This paper is a clean, elegant and logical next step in an important research direction. It feels to me almost like a book chapter. There is only 1 experiment demonstrating that theoretically computed GPKs correspond to practical values. I think you should include experiments that cover all the layer types your theory covers. I would include LinComb+ in the main paper. This is a significant extension that I believe readers should be aware of as it greatly expands the design space for new layer types. I think Corrollary 5.3 is unnecessary. At this point in the paper, it is obvious that you express neural networks of standard architecture as Netsor programs. You don't need to spend a third of a page to reiterate this fact. In fact, I would get rid of section 5 altogether and put all the theory into section 7. I think section 4 and 6 should methodically educate readers on how to a) translate their network into netsor and b) compute the GPK given pairs of inputs and a netsor program. Equation (2) should be included in there as well. I think you should include the Netsor "implementation" of as many building blocks as possible in the main paper, not just batchnorm / convolutions. I believe those transformations are actually the most important contribution of your paper. I also wouldn't title section 6 "More examples" as I think its a severe understatement. The transformations are not just examples, but important insights. (To avoid repeating myself, I also refer to my comments on the NTK paper. Since I wrote that review first, it is a bit more extensive but many suggestions also apply to this paper.) ++++++++ Part 2 ++++++++++++++ I believe that this paper and "Wide Feedforward or Recurrent Neural Networks of Any Architecture are Gaussian Processes" are close to being dual submissions. I believe the policy with regards to similar papers by the same author is that each submission must have substantial independent contributions of the other. Formally, let A and B be papers of the same author under submission. Let contribution() be the set of contributions of a paper. Let T be the threshold of contribution magnitude a paper needs to meet to be accepted. Then as far as I can tell the dual submission policy states that paper A can only be accepted if |contribution(A) \ contribution(B)| > T and paper B can only be accepted if |contribution(B) \ contribution(A)| > T. Contributions that are unique to a single paper are: - the class of nonlinearities used / the presence of matrix transpose in Netsor - the experiments presented (However, these experiments are very small in number and also substantially similar.) I believe that this does not meet the threshold. So based on the standard as outlined above, I would reject both papers. However, I somewhat disagree with the dual submission policy as stated. I believe that paper A should still be accepted if |contribution(A)| > 2T and paper B should still be accepted if |contribution(B)| > 2T, as long as the duplication of material is not gratuitous. And I believe that both papers in this case meet that stadard narrowly. Hence, I will give 6 ratings to both papers. Finally, let me add that I believe the authors could have done a much better job at adding unique content to each paper. NeurIPS enables authors to submit companion papers in the supplementary material. All the content in the NTK paper that is duplicate in the GP paper could have been summarized as "previous work" that reviewers can access in the companion paper. This would have left space open for unique contributions, such as a much larger number of experiments. ############ Post-rebuttal comments ########## Based on reading the other reviews, a consensus has emerged that the NTK paper should be "punished" for overlaps with the GP paper but that the GP paper should be evaluated based on its own merit. Using this standard and taking into account experiments given in the rebuttal, I change my score to an 8.

Reviewer 2



NETSOR gives us a unified perspective on Gaussian process (GP) on various wide neural networks. I think that the framework of NETSOR itself is sufficiently worth to be published. However, I have still several concerns on GP of (a simple) RNN and recommend authors to more clarify their contribution on it. While feedforward neural networks (FNNs) have different weight matrices layer by layer, RNN uses the same weight matrix throughout the whole of time steps. In that sense, FNN and RNN are totally different architectures and it will be highly non-trivial that the RNN also works as GP. Therefore, it will be better to deal with the topic of RNN more carefully. - I recommend Authors to move Section C in Supplementary Material to Section 5 because of the non-trivial property of RNN’s kernel. When I first read the paper, I was confused whether the following two networks have the same kernel or not; (i) a simple RNN with (Ws^1, Ws^2, … Ws^{t-1}) in NETSOR program 2 and (ii) an alternative network with (W_1 s^1, W_2 s^2, W_{t-1} s^{t-1}) where W_k (k=1,…,t-1) are independently given as a fresh i.i.d. copy of W. It should be better to show Section C in the main text and clarify that the RNN has essentially different kernel from the network (ii) because of sharing the same weight throughout the time steps. - What happens in simulations of a simple RNN when the time step t (the length of input sequences k) is large? When the time step t is large, dynamics of generic RNNs relaxed to various equilibrium states; convergence to a fixed point, oscillation or chaotic state. It is a natural question whether GP of a simple kernel has any suggestion on such equilibrium states or irrelevant to them. - RNN includes repeated multiplication of the same weight matrix (e.x. in the case of linear activation function, (W)^t s^1 appear after t time steps) and this is a substantially different point from the previous works on FNNs [13, 29, 31]. As briefly discussed in lines 79—85, Gaussian conditioning technique can overcome this problem. However, the proof of Lem. 7 is written in a very general style, and it is not so easy to imagine how the Gaussian conditioning deals with the repeated multiplication terms in a simple RNN. It should be better to add a brief overview of Gaussian conditioning specific to the simple RNN in Section D. --- After the rebuttal --- I keep my score because Authors have sufficiently responded to other Reviewer's comments.

Reviewer 3



---- Added after discussion After discussion with the reviewers that strongly support publication of one of the two submitted papers, we have reached a consensus that the NTK paper should be incorporated into the GP paper. The strength of the combined papers and the originality and substance of the Gaussian conditioning trick have accordingly led me to raise my score to a weak acceptance. However, I am unwilling to strongly support publication because the following items remain unsatisfactory 1) the utility of NETSOR has still not been adequately articulated and 2) the scope of the proof has not been clearly distinguished from the proofs in Lee et al and de G. Matthews et al. However, with the author's rebuttal and discussions with the other reviewers, I have been convinced that ultimately the strength of the contribution is in extending the GP-NN correspondence even if there is not substantially new theoretical content. ---- This submission focuses on establishing a correspondence between neural networks at initialization (of essentially arbitrary architecture) and Gaussian processes. The primary claim is that any neural network instantiated as a program in the specification language NETSOR introduced by the authors converges to a Gaussian process in the following sense: given Gaussian iid inputs and Gaussian iid initialization the output vector of the NETSOR program converges in distribution to a centered multivariate Gaussian distribution. The result is established using two principal technical tools: first, the NETSOR language itself introduced and used to specify the class of random variables involved; second, a Gaussian conditioning trick originally used in the context of establishing convergence of the TAP equations from spin glass theory is employed to identify the form of the underlying distribution. While I found the general approach of the paper interesting, I have serious concerns about its originality and clarity. What is more, assessments of previous works on the NN-GP correspondence have questioned its significance within the greater context of machine learning because these results hold only at initialization and typically do not provide practical guidance for optimization or training. *Originality* The manuscript thoroughly cites previous work on the topic, but, as far as I can tell, fails to go beyond the results in Refs. 28 and 29. Lee et al. establishes the NN-GP correspondence for arbitrary fully connected deep neural networks using a straightforward induction argument. Matthews et al. arrives at the same result but further provides rates of convergence for the limit in the case that the width of each layer grows at different rates. The present paper aims to include a broader class of networks. In fact, the inductive proofs are only provided for linear combinations and matrix multiplications. Appendices A and B argue that all the other architectures can be decomposed into variables that result from these two operations, meaning that the previous work should also encompass these cases. *Clarity* One of the main contributions of this submission is its NETSOR language for specifying the architecture of a neural network. Basically, this ends up being a scheme for expressing 1) Gaussian random matrices as "A-vars" 2) Gaussian random vectors as "G-vars" 3) the image of Gaussian random vectors as "H-vars". Both conceptually and also in the proofs I found essentially no advantage to using this scheme. It is typically clear that a given weight matrix is a Gaussian random matrix at initialization, so we do not need another name for it. I found that the use of these names for the random variables in the context of the NETSOR programs did more to obfuscate than to aid the proofs. The inductive proof by the Gaussian conditioning trick, on the other hand, is clearly written and explicit. *Significance* I am not sure that the present results are significant: unlike previous work on this topic, the authors do not seek to motivate the correspondence through an explicit connection to training using GP as a prior. I do not believe that it is currently common to use the exact marginal likelihood afforded by the GP to do, e.g., hyperparameter selection, though this may be a route to making the present result actionable. Because the results are for networks at initialization, the practical applications are severely limited in my opinion. *Minor comments* 54: What is the utility of illustrating multi-dimensional output? Isn't it obvious what this means? 79: It would be very useful here to explain in words the conditioning trick and its consequence. 130: give the technical reason for using controlled functions 132: trained neural networks are not Gaussian, so there's no a priori guarantee that the controlled function will be integrable against the limiting parameter distribution after training 206: Why is the output scaling done in this way?

[Author Response · NeurIPS 2019]



*(A)* erf , order vs chaos

*(B)* BN+relu fc net correlations (theory)

*(C)* GRU correlations (theory)

*(D)* transformer correlation (theory)

*(E)* relative deviation from theory

[Meta-Review · NeurIPS 2019]

The paper presents a method for collapsing a wide range of operations (convolution, pooling, batchnorm, attention, gating, as well as the inner products for the actual GP Kernel computation) into the matrix multiplication / nonlinearity / linear combination framework; and also a mean field theory of tied weights, which allows a rigorous extension to RNNs as well as a rigorous integration of the forward and backward pass. The results are novel and interesting. This paper had strong overlap with another paper (that was clearly identified by the authors in both submissions), and so the discussion of the tw o papers took place together.